# The Concordance of Secondary Pathogenic Germline Variants Identified by Tumor Genomic Profiling in Adult Solid Tumor Patients at Two US Community Cancer Centers

**DOI:** 10.3390/genes16121476

**Published:** 2025-12-09

**Authors:** Sarah Moncado, Sourat Darabi, Diana Ivankovic, Luigi Boccuto

**Affiliations:** 1McGlinn Cancer Institute, Tower Health-Reading Hospital, West Reading, PA 19611, USA; 2Hoag Family Cancer Institute, Hoag Presbyterian Hospital, Newport Beach, CA 92663, USA; sourat.darabi@hoag.org; 3Healthcare Genetics and Genomics, Clemson University, Clemson, SC 29634, USA; divanko@clemson.edu (D.I.); lboccut@clemson.edu (L.B.)

**Keywords:** secondary pathogenic/likely pathogenic germline variants, tumor genomic profiling, community cancer care

## Abstract

Background: Secondary pathogenic/likely pathogenic germline variants (P/LPGVs) identified on solid tumor genomic profiling (TGP) are a commonly encountered clinical issue. A proportion of oncology patients that undergo TGP will have a secondary P/LPGV identified that may not have been otherwise discovered based on clinical and family history criteria for hereditary cancer syndrome screening. The confirmation of P/LPGVs on germline sequencing has potential treatment implications for patients. Methods: The study design was a retrospective review for secondary data analysis. The inclusion criteria for this study were adult patients with solid tumor malignancy who underwent TGP and germline sequencing. The objective of this study is to evaluate the concordance rate of secondary P/LPGVs on TGP of adult patients with solid tumor malignancy at Hoag Presbyterian Hospital and Tower Health-Reading Hospital. The second and third aims are to analyze if the confirmed P/LPGVs are concordant with the patient’s tumor type and to analyze the variant allele frequencies (VAFs) of the identified secondary P/LPGVs on the tumor genomic profiling. Results: The data included 75 patients who underwent both TGP and germline sequencing, with a median age of 62.5 years. The most represented genes with P/LPGVs in the combined data included *BRCA1* and *BRCA2*, both with 14, and *MSH2*, with 9. The overall germline concordance rate for the combined population was 64.1%, with 59 out of 92 P/LPGVs identified on both germline and somatic tumor testing. Conclusions: The overall germline concordance rate of 64% for the combined population is in accordance with the reported literature. Possible reasons for the variability in rates could be related to reporting guidelines for secondary germline variants, which can vary by company, and differences between somatic and germline variant curation. The study of P/LPGVs in populations from community cancer centers has the potential to increase the data of underrepresented minority groups regarding this important clinical issue and help expand understanding of hereditary cancer syndrome phenotypes.

## 1. Introduction

Tumor DNA sequencing has excelled at the primary objective of identifying targeted treatments over the past decade with the rise of precision oncology. An increasingly recognized benefit of tumor genomic profiling is the possibility to detect secondary pathogenic germline variants or likely pathogenic germline variants, especially in patients that may not qualify for hereditary cancer syndrome (HCS) screening based on their tumor type and personal and/or family histories. However, a complicating factor with TGP is that confirmatory germline specimens are not required to be sent with the submitted tumor tissue [1]. Often, laboratories only test the tumor sample, and though it is encouraged to report possible P/LPGVs, it is not mandated. To accurately classify a suspected P/LPGV found on tumor-only sequencing, follow-up testing is at the discretion of the ordering oncologist and/or the cancer genetics team.

In a scoping review by Moncado et al. (in press), most of the published literature on secondary P/LPGVs is from larger academic centers or from large biobanks like The Cancer Genome Atlas (TCGA) [2]. The demographics from TCGA included a population where 38% were of Western European descent, 48% did not have a reported race/ethnicity, 5.8% were African American, 3% were Asian, and 4.4% were Hispanic [3].

An identified gap in the literature is the prevalence of secondary P/LPGVs on tumor sequencing at regional or community cancer centers. Approximately 85% of oncology patients receive cancer care in their community [4]. Evaluation of the concordance rate of P/LPGVs at regional cancer centers may help clarify this rate in minority racial/ethnic groups that have been underrepresented in the established literature. Another consideration is that the reported prevalence rates of P/LPGVs in the literature may be affected by socioeconomic factors of the patient populations at tertiary academic centers, such as income, transportation, social support, and physical fitness for medical travel.

The primary objective of this study is to analyze the concordance rate of secondary P/LPGVs on TGP of adult patients with solid tumor malignancy at Hoag Presbyterian Hospital and Tower Health-Reading Hospital. The first aim of the study is to measure the concordance rate of secondary P/LPGVs on tumor genomic profiling in patients with solid tumor malignancy who have also undergone confirmatory germline testing. The second aim is to analyze if the confirmed P/LPGVs are concordant with the patient’s tumor type. The third aim is to analyze the variant allele frequencies (VAFs) of the identified secondary P/LPGVs on the tumor genomic profiling.

## 2. Materials and Methods

The study design is a retrospective review for secondary data analysis. Two regional cancer centers in the United States were selected for this study: Hoag Presbyterian Hospital in Newport Beach, CA, and Reading Hospital in West Reading, PA. For the data from the two community cancer centers, IRB approvals were obtained from Tower Health and Clemson University (Tower Health: 2024-021, Clemson: IRB-2024-0111). Data use agreements were enacted between the healthcare systems and Clemson University. Both IRB approvals had exempt status, and a waiver was granted for informed consent of subjects.

The inclusion criteria for this study were adult patients with solid tumor malignancy who underwent tumor genomic sequencing and hereditary germline sequencing that identified a P/LPGV in a cancer susceptibility gene. For the Hoag Presbyterian Hospital dataset, the period for inclusion was the completion of the tumor genomic profiling between 1 July 2018 and 31 December 2022. For the Reading Hospital dataset, the period for inclusion was the completion of tumor genomic profiling between 1 January 2019 and 31 December 2022. The exclusion criterion was adult subjects with hematologic malignancy.

The pertinent tumor genomic profiling and germline data for each subject were abstracted from the electronic health record, deidentified of patient identifiers, and codified for data analysis. Demographic data was collected, including sex and age; with the Reading Hospital cohort, race/ethnicity was recorded for each participant. The concordance rate was calculated for both health systems, individually and in aggregate. Statistical analyses were facilitated by Excel; all calculations were performed by the first author and verified for accuracy. The concordance rate for P/LPGVs can be defined as the proportion of P/LPGVs identified on tumor genomic profiling and confirmed on germline sequencing. Secondary analysis of the concordance of the P/LPGVs with tumor type was performed at the patient level. Variants of unknown significance were excluded from the concordance rate calculations. Descriptive statistical analysis was performed using demographic data, tumor type, germline gene variants, and variant allele frequency for both health systems individually and in aggregate.

## 3. Results

Reading Hospital and Hoag Presbyterian Hospital data were combined after individual analysis of each institution. Between the two institutions, 75 patients had both tumor genomic profiling and germline sequencing that identified a P/LPGV in a cancer susceptibility gene, with 54 females and 21 males and a median age of 62.5 years (Table 1). The most represented tumor types in the combined data included ovarian, breast, and pancreatic, with 23, 10, and 9 cases, respectively (Figure 1). The most represented genes carrying P/LPGVs in the combined data included *BRCA1* and *BRCA2*, both with 14, and *MSH2*, with 9 (Figure 2 and Figure 3). The overall germline concordance rate for the combined populations was 64.1%, with 59 out of 92 P/LPGVs identified on both germline and somatic tumor testing (95% CI [0.54, 0.74]) (Table 2). There was about equal distribution of somatic VAFs between 40–60% and VAFs > 60%, at 47% and 44%, respectively (Figure 4). Though the phenotype for patients with HCS is typically associated with a younger age of onset, 15 patients across the P/LPGVs were over 70 years old, with 5 carrying variants in *BRCA2*, 4 each in *BRCA1* and *ATM*, and 1 each in *MUTYH* and *MSH2*.

The overall germline concordance rate for the Hoag Presbyterian Hospital population was 67.2%, with 43 out of 64 P/LPGVs identified on the tumor genomic profiling and confirmed with germline sequencing (95% CI [0.55, 0.79]) (Appendix A). The most represented tumor types in the Hoag Presbyterian Hospital dataset were ovarian, breast, and brain, with 22, 8, and 5 cases, respectively (Appendix A). The most represented P/LPGV-carrying genes in the Hoag Presbyterian Hospital dataset were *BRCA1*, *BRCA2*, and *ATM*, with 14, 12, and 6 variants, respectively (Appendix A). The majority of the reported VAFs, about 55%, were between 40-60%, as expected, with a pathogenic variant of germline origin. There was, however, a significant amount of VAFs > 60%, about 43% of the reported P/LPGVs (Appendix A).

The overall germline concordance rate for the Reading Hospital population was 57.1%, with 16 out of 28 P/LPGVs identified on the tumor genomic profiling and confirmed with germline sequencing (95% CI [0.39, 0.75]) (Appendix A). The most represented tumor types in the Reading Hospital dataset were colorectal, pancreatic, and non-small cell lung cancer (NSCLC), with 6, 5, and 3 cases, respectively (Appendix A). The most represented genes with P/LPGVs in the Reading Hospital dataset were *MSH2*, *MUTYH*, and *BRCA2*, with 7, 5, and 3 variants, respectively. Most confirmed P/LPGVS variant allele frequencies were greater than 60%, which is an unanticipated finding as VAFs are typically between 40 and 60% with P/LPGVs (Appendix A).

The most concordant cancer susceptibility genes represented in the combined data were *BRCA1*, *BRCA2* and *ATM*, with 13, 11, and 6 pathogenic/likely pathogenic germline variants, respectively. Expectedly, nine patients with ovarian cancer and one patient with breast cancer had a *BRCA1* pathogenic variant. There were two cases with uncommon genotype–phenotypes, with one patient with brain cancer and another with upper gastrointestinal malignancy. Out of the 15 P/LPGVs in *BRCA1* identified on germline sequencing, 13 were detected on the tumor sequencing profile, resulting in a concordance rate of 86.7% (Figure 2 and Figure 3).

In a patient affected with ovarian cancer, tumor genomic profiling did not identify the P/LPGVS in *BRCA1*; the variant was identified on germline testing. This case also had a P/LPGV in *SDHC* that was identified on the tumor profiling. However, ovarian cancer is an atypical cancer phenotype with *SDHC*, which is typically associated with a risk for pheochromocytoma and paraganglioma [5]. One ovarian cancer case also had a P/LPGV in *TP53*; while ovarian cancer is not one of the most common phenotypes for *TP53*, there can be an association. Another patient with ovarian cancer also had an *FANCC* P/LPGV identified on the germline sequencing only; however, the typical cancer phenotypes associated with *FANCC* variants include head and neck as well as lung squamous cell carcinomas [6]. One patient with NSCLC had a *BRCA1* P/LPGV not identified on the TGP.

Out of the 11 patients with concordant P/LPGVs in *BRCA2*, 3 had variants identified in ovarian cancer, 4 in breast cancer, and 1 each in brain, uterine, pancreatic, and upper GI malignancies. The *BRCA2* P/LPGVs in brain, uterine, and upper GI malignancy are unexpected with these tumor types. The concordance rate for *BRCA2* variants was 68.8%, with 11 out of 16 P/LPGVs identified on tumor genomic profiling (Figure 2 and Figure 3). There was one patient with breast cancer who only had an immunohistochemistry assay performed on the tumor, preventing evaluation for concordance with TGP. There was one patient with ovarian cancer harboring an *MSH3* variant in addition to the *BRCA2* variant, with the former variant also possibly contributing to the phenotype [7]. A case of non-small cell lung cancer had a *BRCA2* variant detected but reported as a VUS on the TGP. Though NSCLC is not typically associated with P/LPGVs in *BRCA2*, there is emerging data that P/LPGVs in homologous recombination repair genes can be seen in patients with NSCLC with younger onset of disease (<60 years old) and without a traditional risk factor like smoking [8].

Six concordant P/LPGVs in *ATM* were identified in three patients with ovarian cancer and one each with pancreatic, thyroid, and non-small cell lung cancers, with the latter tumor type being a less common phenotype [9]. A pancreatic cancer case had insufficient tissue for TGP, so the *ATM* variant could not be confirmed on the somatic sample. The concordance rate for *ATM* was 75%, or six out of eight patients (Figure 2 and Figure 3).

Out of the five patients with concordant P/LPGVs in *MSH2*, two cases were identified in colorectal cancer, two in small bowel cancer, and one in brain cancer. The concordance rate for *MSH2* was 62.5%, with five out of eight P/LPGVs identified on TGP (Figure 2 and Figure 3). The three *MSH2* P/LPGVs not detected on the tumor genomic profiling were due to two cases that were indeterminate on TGP and one case with only immunohistochemistry. All the cases had an expected cancer phenotype of *MSH2* P/LPGVs, including the case of brain cancer, as susceptibility to brain tumors, mainly glioblastoma, is associated with Lynch syndrome [7].

The four concordant *MUTYH* P/LPGVs were identified in two cases of colorectal cancer and one each of ovarian and non-small lung cancers, with the latter two being uncommon cancer phenotypes [10]. The concordance rate was 80%, with one *MUTYH* P/LPGV in a pancreatic cancer case that was not detected on the tumor genomic profiling. There were also four concordant P/LPGVs identified in *CHEK2*, with one case each in breast, pancreatic, brain, and prostate cancers (Figure 2 and Figure 3). While prostate cancer is an expected phenotype of *CHEK2*, brain cancer is not [11]. The brain cancer case also had a P/LPGV in *BRCA1*, which is also a discordant phenotype. The prostate cancer case also had P/LPGVs in *BARD1*, *NF2*, and *POLE*, all of which were not reported on the TGP. The concordance rate for the *CHEK2* P/LPGVs was 100%.

All three P/LPGVs identified in *APC* were concordant, with one each in sarcoma and ovarian and brain cancers. Though patients with *APC* germline variants are typically at risk for colorectal cancers, extracolonic malignancies like medullary thyroid cancer and medulloblastomas are included in the syndrome [12]. For this study, the pathologic subtypes of thyroid and brain cancers were not included in the data extraction. The *FH* cancer susceptibility gene also had three concordant P/LPGVs, with one each in brain, renal, and upper GI cancers, with just renal cell carcinoma being a common phenotype. The concordance rate was 75%, as the tumor genomic profiling in a case of uterine cancer did not identify the *FH* P/LPGV (Figure 2 and Figure 3). The patient with uterine cancer also had *FLCN*, *MEN1* and *NF1* PGVs, though none of these three PGVs are typically associated with the uterine carcinoma phenotype either [13,14]. P/LPGVs with one case each are summarized in Table 2.

**Table 2 genes-16-01476-t002:** Pathogenic/likely pathogenic germline variants on germline testing stratified by patient.

Gene(s)	Tumor Type	Variant(s)	Detected on TGP?	Common Genotype–Phenotype Association?
** *APC* **	Ovarian	NM_000038.6:c.3920T>A	No	No
** *APC* **	Sarcoma	NM_000038.6:c.3920T>A	Yes	No
** *APC* ** ** *FH* **	Brain	NM_000038.6:c.3920T>A NM_000143.4:c.1431_1433dup	*APC/FH*: Yes	No No
** *ATM* **	Pancreatic	NM_000051.3:c.3448A>T	NGS failed	Yes
** *ATM* ** ** *BRCA2* **	NSCLC	NM_000051.3:c.7271T>G NM_000059.3:c.619A>G	*ATM*: Yes *BRCA2*: VUS	No No
** *ATM* **	Ovarian	NM_000051.4:c.8418+5_8418+8del	Yes	Yes
** *ATM* **	Pancreatic	EX62_63del	No	Yes
** *ATM* **	Ovarian	NM_000051.4:c.2921+1G>A	Yes	Yes
** *ATM* **	Thyroid	NM_000051.4:c.7517_7520del	Yes	Yes
** *ATM* **	Pancreatic	NM_000051.4:c.2502dup	Yes	Yes
** *BARD1* **	Breast	NM_000465.4:c.1996C>T	No	Yes [15,16]
** *BRCA1* **	NSCLC	NM_007294.3:c.5213_5278-2753del	No	No
** *BRCA1* **	Breast	NM_007294.4:c.2681_2682del	Yes	Yes
** *BRCA1* **	Ovarian	NM_007294.4:c.4524G>A	Yes	Yes
** *BRCA1* **	Ovarian	NM_007294.4:c.3964A>T	Yes	Yes
** *BRCA1* **	Other	NM_007294.4:c.178C>T	Yes	Unknown
** *BRCA1* **	Ovarian	NM_007294.4:c.68_69del	Yes	Yes
** *BRCA1* **	Ovarian	NM_007294.4:c.68_69del	Yes	Yes
** *BRCA1* ** ** *CHEK2* **	Brain	NM_007294.4:c.68_69del NM_007194.4:c.1100del	*BRCA1*, *CHEK2*: Yes	No No
** *BRCA1* **	Ovarian	NM_007294.4:c.3759_3760del	Yes	Yes
** *BRCA1* **	Ovarian	NM_007294.4:c.962G>A	Yes	Yes
** *BRCA1* ** ** *TP53* **	Ovarian	NM_007294.4:c.3749_3752del NM_000546.6:c.727A>G	*BRCA1:* Yes *TP53:* No	Yes No
** *BRCA1* **	Upper GI	NM_007294.4:c.131G>T	Yes	No
** *BRCA1* ** ** *FANCC* **	Ovarian	NM_007294.4:c.329dup NM_000136.3:c.770T>C	*BRCA1*: Yes *FANCC:* No	Yes No
** *BRCA1* ** ** *SDHC* **	Ovarian	NM_007294.4:c.4096+1G>A NM_003001.5:c.43C>T	*BRCA1:* No *SDHC:* Yes	Yes No
** *BRCA2* **	Breast	NM_000059.3:c.3458delA	Yes	Yes
** *BRCA2* **	Pancreatic	NM_000059.4:c.5350_5351delAA	Yes	Yes
** *BRCA2* **	Breast	NM_000059.3:c.2808_2811delACAA	Yes	Yes
** *BRCA2* **	Ovarian	NM_000059.4:c.5197_5198delTC	Yes	Yes
** *BRCA2* **	Ovarian	NM_000059.4:c.5645C>A	Yes	Yes
** *BRCA2* **	Breast	NM_000059.4:c.4478_4481del	Yes	Yes
** *BRCA2* **	Brain	NM_000059.4:c.5946delT	Yes	No
** *BRCA2* **	Breast	NM_000059.4:c.145G>T	Yes	Yes
** *BRCA2* **	Uterine	NM_000059.4:c.7480C>T	Yes	No
** *BRCA2* **	Breast	NM_000059.4:c.3264dup	No *	Yes
** *BRCA2* **	Upper GI	NM_000059.4:c.6724_6725del	Yes	No
** *BRCA2* ** ** *ATM* **	Ovarian	NM_000059.4:c.1399A>T NM_000051.4:c.6100C>T	*BRCA2*, *ATM*: Yes	Yes Yes
** *BRCA2* **	Ovarian	5′UTR_EX1del	No	Yes
** *BRCA2* ** ** *MSH3* **	Ovarian	EX13_15del NM_002439.5:c.2647G>T	*BRCA2*: No *MSH3*: Yes	Yes Yes
** *BRCA2* **	Breast	NM_000059.4:c.4631del	No *	Yes
** *BRIP1* **	Ovarian	NM_032043.3:c.1343G>A	No *	Yes [15]
** *CHEK2* ** ** *MLH1* ** ** *NF1* **	Pancreatic	NM_007194.4:c.470T>C NM_000249.4:c.1633A>G NM_001042492.3:c.367A>G	*CHEK2*: Yes *MLH1*, *NF1*: No	Yes Yes No
** *CHEK2* ** ** *BARD1* ** ** *NF2* ** ** *POLE* **	Prostate	NM_007194.4:c.1100del NM_000465.4:c.1403C>T NM_000268.4:c.1490G>C NM_006231.4:c.14G>A	*CHEK2:* Yes; other P/LPGVs: No	Yes No [15,16] No No
** *CHEK2* **	Breast	NM_007194.4:c.349A>G	Yes	Yes
** *FH* **	Upper GI	NM_000143.4:c.1431_1433dup	Yes	No
** *FH* ** ** *NTHL1* **	Renal	NM_000143.3:c.1108+1G>T NM_002528.6:c.71G>C	*FH*: Yes *NTHL1*: No	Yes No
** *FH* ** ** *FLCN* ** ** *MEN1* ** ** *NF1* **	Uterine	NM_000143.4:c.194A>G NM_144997.7:c.614T>C NM_001370259.2:c.511C>T NM_001042492.3:c.4294G>A	All P/LPGVs: No	No No No No
** *FLCN* **	Ovarian	NM_144997.7:c.780-3C>A	No	No
** *HOXB13* **	Upper GI	NM_006361.6:c.251G>A	Yes	No [17]
** *LZTR1* **	Spine	NM_006767.4:c.1397G>A	No	No [18]
** *MLH1* **	Colon	NM_000249.4:c.350C>T	Yes	Yes
** *MSH2* **	Brain	NM_000251.3:c.942+3A>T	Yes	Yes
** *MSH2* ** ** *POT1* **	Colon	NM_000251.2:c.942+1G>T NM_015450.2:c.452T>G	*MSH2:* Yes *POT1:* No	Yes Yes
** *MSH2* **	Colon	NM_000251.2:c.2635-?_*279+?del	No	Yes
** *MSH2* **	Uterine	NM_000251.2:c.-125_1076+?del	No	Yes
** *MSH2* **	Small bowel	NM_000251.2:c.943-1G>T	Yes	Yes
** *MSH2* **	Prostate	NM_000251.2:c.942+3A>T	No	No
** *MSH2* **	Colon	NM_000251.3: c.790C>T	Yes	Yes
** *MSH2* **	Small bowel	NM_000251.3:c.2005+1G>A	Yes	Yes
** *MSH6* **	Colon	NM_000179.3:c.1444C>T	Yes	Yes
** *MUTYH* **	NSCLC	NM_001128425.1:c.1187G>A	Yes	No
** *MUTYH* **	Ovarian	NM_001128425.1:c.1187G>A		No
** *MUTYH* **	Pancreatic	NM_001128425.1:c.470C>T	No	Yes
** *MUTYH* ** ** *MUTYH* **	Rectal	NM_001128425.2c.536A>G NM_001048174.2:c.1214C>T	Both *MUTYH*: Yes	Yes Yes
** *PALB2* **	Pancreatic	NM_024675.4:c.509_510delGA	Yes	Yes
** *RAD50* ** ** *SMAD4* **	Upper GI	NM_005732.4:c.1663A>G NM_005359.6:c.424+6T>C	*RAD50*, *SMAD4*: No	No Yes [19]
** *RAD51C* **	Breast	EX4_3′UTRdel	No	Yes
** *RAD51D* **	Colon	NM_002878.4:c.388C>T	No	No [20]
** *PTEN* **	Colon	NM_000314.8:c.131dup	Yes	Yes
** *SPINK1* **	Pancreatic	NM_001379610.1:c.101A>G	No	Yes

Key—TGP: tumor genomic profiling; NGS: next-generation sequencing; IHC: immunohistochemistry. * IHC positive for protein expression of the P/LPGVs but not detected on the NGS.

## 4. Discussion

This study highlights the importance of accurately reporting possible P/LPGVs on tumor genomic profiling, as it can trigger germline testing in patients who may not have met current clinical criteria for hereditary cancer syndrome screening. The overall germline concordance rate of 64% for the combined population is within the range of the published literature. For comparison, a retrospective study published by Pauley et al. in 2022 reported a concordance rate of 82% in a population of 2811 patients with solid tumors from the Huntsman Cancer Institute [21]. A study published by Darabi et al. in 2024 had a concordance rate of 43.6% in 39 patients selected for moderately to highly penetrant genes associated with hereditary cancer syndromes [22]. Some possible explanations for the variability in rates could be related to reporting guidelines for secondary germline variants, which can vary by company, and differences between somatic and germline variant curation. The variability in concordance rate highlights the fact that the primary reason for the sequencing is to find targetable treatments; also, it is not mandatory for sequencing companies to report P/LPGVs. Identifying uncommon tumor phenotypes with P/LPGVs potentially has treatment implications for the patients once germline sequencing confirms the P/LPGV. Confirming P/LPGVs also allows for cascade testing for at-risk family members.

The most represented tumor types with P/LPGVs between the two centers were ovarian, breast, and pancreatic cancers. In the scoping review by Moncado et al. (in press), which evaluated the prevalence of P/LPGVs in solid tumor genomic profiling, breast, ovarian, and colorectal cancers had the highest frequencies of P/LPGVs, followed by pancreatic cancer [2]. The most represented genes harboring P/LPGVs for this study were *BRCA1*, *BRCA2*, and *MSH2*. *BRCA1*, *BRCA2*, and *MUTYH* were the genes with the most frequent P/LPGVs in the scoping review, while *MSH2* was 11th out of 28 hereditary cancer syndrome genes included for analysis [2]. The relative increase in P/LPGVs in *MSH2*, a Lynch syndrome-associated gene, is explainable by the Reading Hospital cohort, with colorectal cancer as the most frequent tumor type. The high prevalence of P/LPGVs in these genes in our study and in the scoping review is expected, as these are all tumor suppressor genes associated with hereditary cancer syndromes. Four founder variants were identified in the Hoag Presbyterian Hospital cohort: three involved the *BRCA1* c.185delAG variant and one *BRCA2* c.4859delA variant [23]. There were no founder variants in *BRCA1* or *BRCA2* in the Reading Hospital cohort.

The VAFs of the combined data were almost equally distributed between 40–60% and >60%. P/LPGVs in cancer predisposition genes are expected to have a VAF between 40 and 60%; however, the increased frequency of pathogenic germline variants with VAFs > 60% can potentially be explained by loss of heterozygosity in the tumor. Loss of heterozygosity is defined by the loss of the wild-type allele, which allows the pathogenic variant to be overexpressed in the tumor sequencing [24]. Other explanations include allelic imbalance from copy number variation, leading to lower or higher expression of one allele at the gene locus, and low tumor purity estimation, resulting in inaccurate VAF calculation.

The racial/ethnicity demographics of the Reading Hospital cohort did skew White, which is expected since about 68% of the Berks County population is White according to the 2020 Census [25]. However, about 25% of the county is Hispanic, with most of this population living in Reading, PA, where the cancer center is located [25]. The Hispanic population in the Reading Hospital cohort was only 9%, or 2 patients out of 22. This finding can be contextualized by a study evaluating Medicare patients with gastrointestinal, lung, and breast cancers diagnosed from 2015 to 2020. The study cohort had 1.4 million patients, with only 1.8% having tumor genomic profiling. About 89% of the patients who had tumor genomic profiling were White, in comparison to 0.5% of Hispanic patients [26]. The entire Hispanic population in the study was 1.4%, emphasizing the lack of representation in a large healthcare payer system like Medicare.

There were limitations to this study design. This study was a retrospective review over an approximately 3-year period with a selection bias of patients who had both positive germline testing for a P/LPGV and tumor genomic profiling. This limited the accrual to 75 patients between both institutions. Another limitation is that the true prevalence of pathogenic/likely pathogenic germline variants could not be calculated due to the design of this study. Missing somatic variant data on some of the tumor specimens affected the data analysis. Low numbers of P/LPGVs by tumor type and hereditary cancer syndrome gene precluded statistical subanalysis. The generalization of the data is limited due to the small numbers in this study and the unique population makeup of the two community cancer centers.

Future directions of this research include the evaluation of the secondary pathogenic germline variants in a larger cohort through more aggregation of patients receiving cancer care in community centers. Another way to increase the cohort would be to increase the period for review, but the rapidity of change of included somatic gene variants on tumor profiling would likely be a significant confounder. It is important to evaluate patients in community cancer centers as there is an opportunity to include underrepresented populations in research. Another extension of this study is evaluating the prevalence of secondary germline variants in patients receiving cancer care in their community to compare to the published literature from tertiary centers and biobanks.

The potential to identify incidental pathogenic germline variants in patients who may not meet current guidelines for hereditary cancer syndrome screening has important implications for both patients and their families. Another strategy to capture patients who may not meet hereditary cancer syndrome screening requirements is to send tumor-normal specimens to perform both somatic and germline testing simultaneously.

## 5. Conclusions

Secondary pathogenic germline variants on tumor sequencing are well-recognized as a clinical issue in oncology care. Our study evaluated the concordance rate between germline sequencing and tumor genomic profiling in patients receiving care in two community cancer centers, with an overall rate of 64%. This highlights the imperative to improve standardization of the inclusion, identification, and reporting of gene variants that have a high likelihood of germline origin on tumor genomic profiling across tumor sequencing companies. Routine use of tumor-normal specimen testing would help to better define the prevalence of pathogenic germline variants in patients with cancer. This approach has the potential to further expand our understanding and recommendations for screening of hereditary cancer syndromes, which would benefit the whole human population.

## Figures and Tables

**Figure 1 genes-16-01476-f001:**
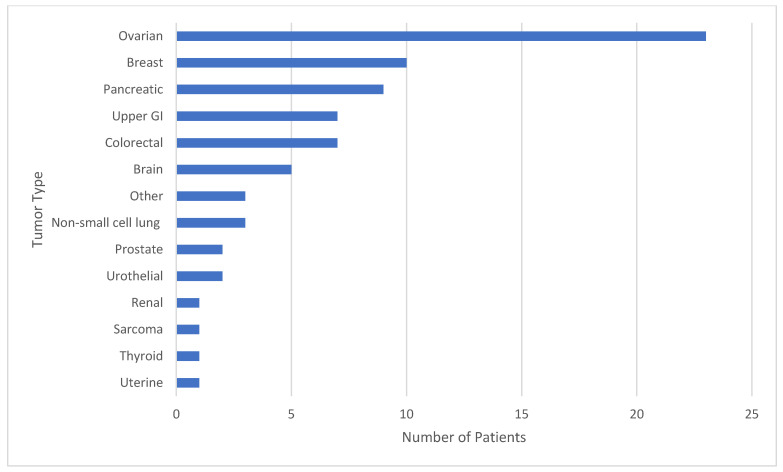
The study population by tumor type. The number of patients from Hoag Presbyterian Hospital and Reading Hospital represented by tumor type. The most represented tumor types in the study population included ovarian, breast, and pancreatic cancers, expectedly, as these tumor types have strong hereditary cancer syndrome associations.

**Figure 2 genes-16-01476-f002:**
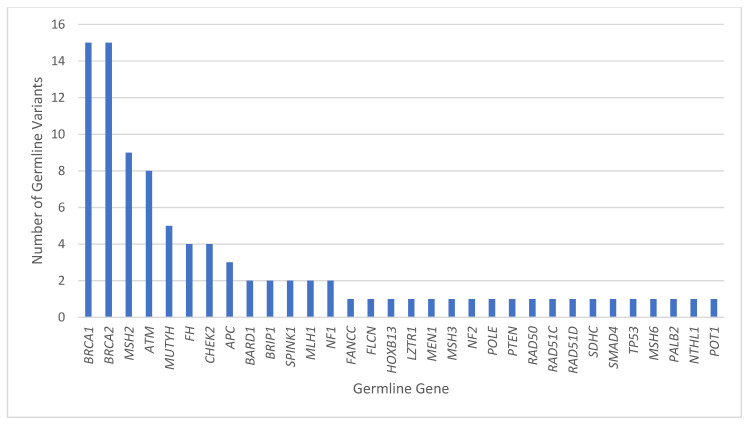
Combined pathogenic/likely pathogenic germline variants by gene. The number of P/LPGVs identified on germline sequencing by hereditary cancer syndrome gene from Hoag Presbyterian Hospital and Reading Hospital. The genes with the most P/LPGVs identified were *BRCA1*, *BRCA2*, *MSH2*, and *ATM*, which correlates phenotypically with our top 5 malignancies, including ovarian, breast, upper GI, pancreatic, and colorectal cancers.

**Figure 3 genes-16-01476-f003:**
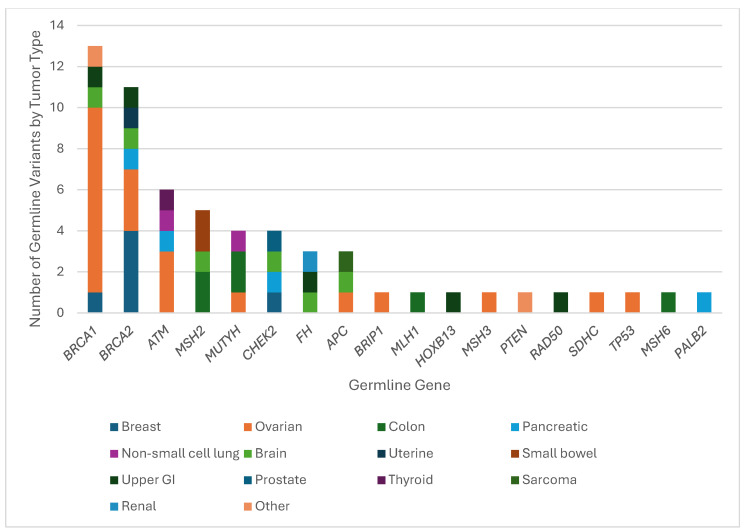
Hereditary cancer syndrome genes with pathogenic/likely pathogenic germline variants by tumor type. The number of confirmed P/LPGVs from TGP stratified by hereditary cancer syndrome gene and tumor type from Hoag Presbyterian Hospital and Reading Hospital. The most represented tumor type was ovarian cancer across *BRCA1*, *BRCA2,* and *ATM*.

**Figure 4 genes-16-01476-f004:**
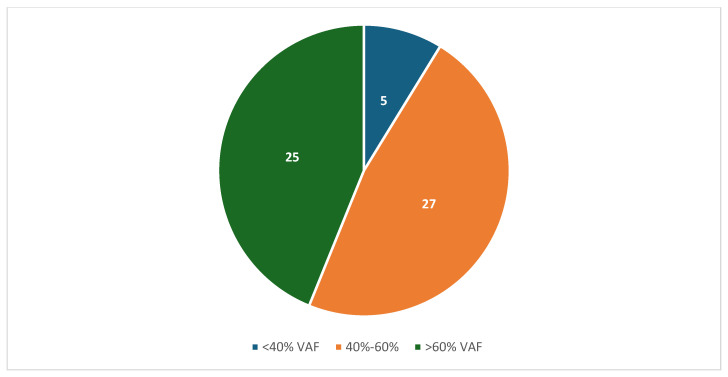
The number of combined variant allele frequencies from TGP of confirmed pathogenic/likely pathogenic germline variants from Hoag Presbyterian Hospital and Reading Hospital. The majority of P/LPGV VAFs were about equal between 40–60% and >60%.

**Table 1 genes-16-01476-t001:** Demographics.

Patient Characteristics	Patients with P/LPGVs
Total (n = 75)	
Age at diagnosis	
30–44 yo	7 (9%)
45–54 yo	6 (8%)
55–64 yo	30 (40%)
65–74 yo	16 (21%)
75–84 yo	13 (17%)
85–100 yo	3 (4%)
Gender	
Male	21 (28%)
Female	54 (72%)
Race/Ethnicity (Reading Hospital only)	
Caucasian	18 (82%)
African American	2 (9%)
Hispanic	2 (9%)

## Data Availability

The datasets presented in this article are not readily available because of patient privacy restrictions.

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
