# Peer review of "The Concordance of Secondary Pathogenic Germline Variants Identified by Tumor Genomic Profiling in Adult Solid Tumor Patients at Two US Community Cancer Centers"

_genes, 2025, doi:10.3390/genes16121476_

Round 1
Reviewer 1 Report
Comments and Suggestions for Authors
One important clinical question that the study aims to address is the frequency with which pathogenic or potentially pathogenic mutations detected by tumor genomic profiling (TGP) are confirmed by germline testing. Although the findings may be useful, there are some significant problems in the text that require modification or more evaluation.
How frequently pathogenic or potentially pathogenic variants identified by tumor genomic profiling (TGP) are validated by specific germline testing is a significant clinical topic that the study attempts to answer. The retrospective review of 75 patients yielded a combined concordance of 64.1%, with BRCA1, BRCA2, and MSH2 the most frequently affected genes and ovarian, breast, and pancreatic cancers the most common tumor types. Variant allele frequencies (VAFs) were reported roughly equally in the 40–60% and >60% bins. The findings are potentially valuable, but the manuscript contains a number of substantive issues that need revision or additional analysis.
The manuscript currently communicates contradictory interpretations of how the combined concordance compares to prior work. In some places the 64% concordance is described as lower than reported literature, while the Discussion describes the same 64% as being within the range of published estimates. This internal contradiction should be resolved.
The definition of “concordance rate” is ambiguous in the manuscript and appears to be used with different denominators in different places. At times the text reads as if the denominator is the variants identified on tumor genomic profiling, while elsewhere the wording implies the denominator is variants found on germline testing. This matters because each denominator answers a different question: using germline-reported variants as the denominator measures how often a known germline P/LP variant is detected on TGP (essentially a sensitivity-like measure), while using tumor-reported variants as the denominator measures the positive predictive value of tumor calls for germline origin.
The unit of analysis is unclear in places. The manuscript reports 92 P/LPGVs across 75 patients, which implies a variant-level analysis because some patients carry more than one variant. However, parts of the text read as though patient-level statistics are being reported (for instance when discussing genotype–phenotype concordance). The Methods and Results should explicitly state that the primary analyses are variant-level, and secondary or complementary summaries should be clearly identified as patient-level.
The technical details of the TGP and germline assays are insufficient. The paper attributes discordance to “variability of gene inclusion and depth of sequencing” across commercial TGP providers, but does not provide the concrete information to evaluate that claim. Please add a Methods subsection that lists, for each TGP provider represented in the cohort (or for each TGP platform if multiple were used), the panel name, the number of genes tested, typical read depth, whether matched tumor-normal sequencing was performed, and the variant-calling pipeline and version/date. For germline testing, report the lab name, the assay type (targeted NGS panel vs. exome), whether CNV/MLPA analysis was performed, and the classification framework used.
The results present point estimates for concordance but do not include uncertainty measures or statistical comparisons. Given the relatively small sample sizes, the precision of the estimates is limited and should be reported. I recommend adding 95% confidence intervals (Wilson or exact binomial) for all reported proportions and, where you compare centers, an appropriate statistical comparison (Fisher’s exact test or chi-square).
The manuscript gives slightly different VAF distributions in different sections (for example, a combined statement that 47% of confirmed P/LPGVs are between 40–60% and 44% are >60% is later contrasted with site-specific descriptions that do not sum consistently). These apparent inconsistencies should be resolved by presenting a single clear table that reports VAF categories (<40%, 40–60%, >60%) with counts and percentages for the combined cohort and for each center separately so that values sum to 100% in each column. Moreover, attributing VAFs >60% to loss of heterozygosity (LOH) is plausible but not defensible without supporting copy-number or allele-specific data and tumor purity estimates. Tumor purity and copy-number changes are alternative explanations for elevated VAFs, so either provide supporting CNV/allelic imbalance evidence if you wish to claim LOH, or soften the language and acknowledge tumor purity/CNV as plausible contributors.
At present the table 2 mixes inconsistent gene labels (e.g., “BRCA” vs “BRCA1”), repeated variant entries (c.68_69del appears multiple times without clear explanation), unclear HGVS notations (for example “MSH2 c.2635-?_*279+?del”), and entries such as “IHC only / Yes” that conflate protein IHC results with sequence variant detection. Use standardized HGVS nomenclature (including transcript accession, e.g., NM_007294.4:c.68_69del), remove unexplained duplicates (or explain duplicates as multiple patients), and add explicit columns such as “Detected on TGP?” (Detected / Not detected / VUS / Indeterminate / NGS failed). Also state clearly how VUS calls were handled in concordance calculations; if VUS were excluded, say so and report the counts of VUS separately.
The Methods and Limitations note that the design selected patients who had both TGP and germline sequencing and appears to have focused on those with positive germline results, which prevents estimation of prevalence and introduces selection bias. To make this transparent, include a patient flow diagram (CONSORT-style) showing the total number of patients considered, how many had TGP only, germline only, both, and how many were excluded (and for what reasons). Explicitly state whether the cohort was restricted only to patients who had a known P/LP germline result.
The manuscript flags several “unexpected” gene–tumor pairings (for example, BRCA2 in brain tumors), but given the variant-level analysis and the selection strategy that enriched for germline positives, the observed frequency of unexpected pairings may be biased. Soften categorical language about “unexpected” associations, frame these observations as hypotheses or “reported but uncommon” associations, and, where possible, cite relevant literature or ClinVar/other databases that document the variant’s prior observations. A short table that lists the apparently discordant pairings and cites supporting literature would be helpful.
Variant classification harmonization is not addressed. Clinical labs can apply ACMG criteria differently and classifications can change over time, so it is important to state whether you used the clinical laboratory’s original calls, or whether the study team re-classified variants using a common standard (for example ACMG 2015). If classifications were not harmonized, acknowledge this as a limitation and consider re-classification for key discordant variants.
Minor revision
There are several small but important editorial fixes that will improve readability and accuracy.
Abbreviations for pathogenic variants are used inconsistently (P/LPGV, PGV, P/LPGVS, etc.); choose one style (for example P/LPGV) and use it consistently throughout the manuscript.
Correct obvious typos and such as “The were limitations to this study design” should be “There were limitations to this study design,”.
Table 2 needs clearer column headings and unambiguous entries;
Verify that all figures and supplemental figures referenced (S1–S7) are included, correctly labeled, and cited at the correct points in the text.
For variant nomenclature, add transcript accession numbers and zygosity (heterozygous/homozygous) where relevant to prevent ambiguity in Table 2 and elsewhere.
State whether VUS were counted as positive, excluded from concordance calculations, or reported separately and include their counts.
Reviewer 2 Report
Comments and Suggestions for Authors
The manuscript addresses an important clinical topic: the concordance of secondary pathogenic/likely pathogenic germline variants (P/LPGVs) identified through tumor genomic profiling (TGP) with confirmed germline testing in adult patients with solid tumors. The study’s focus on community cancer centers is particularly valuable, as these populations are often underrepresented in genomic research. However, the manuscript has several significant issues that limit its clarity and overall suitability for publication in its current form. The presentation of results is overly dense and repetitive, and the introduction and abstract do not provide sufficient context or rationale for the study. The title is vague and does not clearly communicate the study’s population, methods, or significance.
Specific Recommendations:
- The authors should improve the title. For example:
“Concordance of Pathogenic Germline Variants Identified by Tumor Genomic Profiling in Adult Solid Tumor Patients at Two US Community Cancer Centers.” - The abstract needs reorganization and rephrasing to describe the primary objective of the study, the main results of the study and the clinical significance or implications of the findings.
- The authors should improve the introduction section to better explain the clinical challenge posed by secondary P/LPGVs and state study aims and hypotheses.
- The addition of extra tables and/or figures to summarize gene-specific results rather than long narrative paragraphs is necessary.
- Move detailed case-level genotype-phenotype descriptions to a supplemental section.
- Reduce repetition and long, complex sentences throughout the text.
- Emphasize the implications of concordance rates and unexpected genotype-phenotype findings in the discussion.
- Highlight how these data could impact clinical decision-making or patient counseling in community cancer centers.
Reviewer 3 Report
Comments and Suggestions for Authors
Our research colleagues have submitted a very interesting paper. The objectives are to evaluate the concordance rate of pathogenic/likely secondary pathogenic germline variants in the genomic profile of solid tumors from adult patients with solid tumor malignancies. The second and third objectives are to analyze whether the confirmed pathogenic/likely pathogenic germline variants are concordant with the patient's tumor type and to analyze the variant allele frequencies of the pathogenic/likely secondary pathogenic germline variants identified in the tumor genomic profile. The abstract is an excellent summary of the entire study. The next section perfectly introduces the topics covered in the study. The phrase "patients who may not be eligible for screening for hereditary tumor syndrome (HCS) based on tumor type and personal and/or family history" is particularly important. It is appropriate to note that in certain areas of the world, the socioeconomic conditions of populations limit and often influence studies, causing distortions in the data that are then evaluated globally. The final section of the introduction explains the objectives of the study, and the materials and methods explain that the study was retrospectively conducted at two centers where the genomic profiles of hematological tumor cells were studied. Of particular significance is the concept of studying the concordance rate for pathogenic/probably secondary pathogenic germline variants on the genomic profile. This can be defined as the proportion of pathogenic/probably secondary pathogenic germline variants on the genomic profile identified through tumor genomic profiling and confirmed by germline sequencing. The use of Excel is essential for examining the genetic sequence. We find it extremely interesting to find that the most represented cancer types in the combined data at the first hospital included ovarian, breast, and pancreatic cancers with the BRCA1, BRCA2, and MSH2 genes. At the second hospital, these were colorectal, pancreatic, and non-small cell lung cancers with the MSH2, MUTYH, and BRCA2 genes. The results, which should be read with the relevant tables several times, are well described. The paper prepares for conclusions based on the results and the literature, but the bottom line is that secondary pathogenic variants in tumor sequencing are widely recognized as a clinical problem in oncology care. The suggestion that routine testing on normal tumor samples would help better define the prevalence of germline pathogenic variants in cancer patients is commendable. This study would benefit the entire human population if screening for hereditary cancer syndromes were performed. Excellent English, excellent iconography, good literature.
Author Response
The reviewer did not have specific comments to address.
Reviewer 4 Report
Comments and Suggestions for Authors
This retrospective, two-center study by Moncado et al., evaluates concordance of secondary pathogenic/likely pathogenic germline variants (P/LPGVs) detected by tumor genomic profiling (TGP) with confirmatory germline testing in 75 adults with solid tumors. The authors report 92 P/LPGVs on TGP, 59 of which were confirmed on germline testing (64.1% concordance overall). They further describe tumor-type “phenotype concordance” and summarize variant allele frequency (VAF) distributions, noting many confirmed variants with VAF >60%. The topic is timely and clinically relevant—especially the focus on community settings, where data are limited. The manuscript is clearly organized and includes thoughtful phenotype notes. That said, the study would benefit from stronger methods transparency, clearer denominators, basic inferential statistics.
Specific comments:
- It appears concordance is assessed per variant (59/92), not per patient. Please state this explicitly in Methods and keep denominators consistent throughout Results and tables. Consider also presenting per-patient concordance (e.g., “% patients with ≥1 secondary P/LPGV on TGP that was germline-confirmed”) and a patient-level flow diagram.
- Please list the TGP platforms, gene panels, capture methods, mean coverage, and tumor purity estimates (or a proxy). These strongly influence VAF and sensitivity, and are essential to interpret center-level differences and missing calls.
- Table 2 mixes genes, variants, and “TGP Status” entries like “IHC only / Yes.” Please split into clean columns (Gene, Zygosity if known, TGP detection: Yes/No/Not assessable). This will reduce ambiguity and facilitate reanalysis. Where multiple PGVs are listed for a patient, consider using a patient ID so the reader can see co-occurrence and how each variant contributed to the concordance computation.
- Core data not available for this study. I am not sure if this adheres to the journal policy.
Round 2
Reviewer 1 Report
Comments and Suggestions for Authors
The article has been revised based on the suggestions and indications. However, it is necessary to clarify the first point. The abstract conclusion stated, "The combined population's overall germline concordance rate of 64% is lower than in the reported literature," whereas the discussion stated, "The combined population's overall germline concordance rate of 64% is within the range of the published literature." So check better.
The answers, one by one, are appropriate and clear. I do not believe that any other adjustments are necessary.
Author Response
Comment 1: The article has been revised based on the suggestions and indications. However, it is necessary to clarify the first point. The abstract conclusion stated, "The combined population's overall germline concordance rate of 64% is lower than in the reported literature," whereas the discussion stated, "The combined population's overall germline concordance rate of 64% is within the range of the published literature." So check better.
The answers, one by one, are appropriate and clear. I do not believe that any other adjustments are necessary.
Response: Thank you to the reviewer for highlighting this oversight in the abstract. The statement has been updated to reflect the statement in the discussion.
Reviewer 2 Report
Comments and Suggestions for Authors
Despite minor editorial improvements, the manuscript remains primarily descriptive, with limited analytical rigor and no substantial novelty. The methodological simplicity and repetitive structure do not meet Genes’ publication standards for a full research article. A more suitable venue might be a regional or clinical genetics journal focused on community-based oncology data summaries.
Major Concerns
- The research primarily reports descriptive concordance rates (64% overall) without providing new analytical or mechanistic insights. The revisions did not introduce additional validation, statistical modeling, or broader data contextualization.
- Multiple sentences, including entire paragraphs in the Introduction and Methods sections, are repeated (e.g., lines 23–27 and 98–101).
- The authors rely on Excel for statistical analysis, which is inappropriate for handling confidence intervals, concordance calculations, and stratified analyses.
- No statistical comparison between subgroups (e.g., by tumor type or gene) is performed to support the authors’ conclusions.
- The Discussion repeats known findings and lacks critical interpretation of why concordance differs across centers, gene classes, or variant types.
- The manuscript includes numerous typographical and grammatical errors (“is withlower,” “an unanticipated un-common,” “a commonexpected phenotype”).
- Figures and supplementary materials are descriptive but do not provide new insight or resolve the fundamental limitations of the study design.
Minor Issues
- Some citations are redundant or inconsistently formatted.
- figure legends lack sufficient explanation to interpret the data independently.
- The manuscript needs professional language editing.
- The manuscript includes numerous typographical and grammatical errors (“is withlower,” “an unanticipated un-common,” “a commonexpected phenotype”).
Reviewer 4 Report
Comments and Suggestions for Authors
The authors addressed my questions.
Author Response
Comment 1: The authors addressed my questions.
Response: No response required.